# Cosmetic Functional Ingredients from Botanical Sources for Anti-Pollution Skincare Products

**Claudia Juliano *** [iD] **and Giovanni Antonio Magrini**

Department of Chemistry and Pharmacy, University of Sassari, Via Muroni 23/A, 07100 Sassari, Italy;
giovanniantonio.magrini@gmail.com
* Correspondence: julianoc@uniss.it; Tel.: +39-7922-8729

**Abstract:** Air pollution is a rising problem in many metropolitan areas around the world. Airborne contaminants are predominantly derived from anthropogenic activities, and include carbon monoxide, sulfur dioxide, nitrogen oxides, volatile organic compounds, ozone and particulate matter (PM; a mixture of solid and liquid particles of variable size and composition, able to absorb and delivery a large number of pollutants). The exposure to these air pollutants is associated to detrimental effects on human skin, such as premature aging, pigment spot formation, skin rashes and eczema, and can worsen some skin conditions, such as atopic dermatitis. A cosmetic approach to this problem involves the topical application of skincare products containing functional ingredients able to counteract pollution-induced skin damage. Considering that the demand for natural actives is growing in all segments of global cosmetic market, the aim of this review is to describe some commercial cosmetic ingredients obtained from botanical sources able to reduce the impact of air pollutants on human skin with different mechanisms, providing a scientific rationale for their use.

**Keywords:** plants; plant extracts; anti-pollution ingredients; antioxidant

## 1. Introduction

Nowadays air pollution is a global environmental and health problem of growing concern. While some kinds of air pollution are produced naturally (forest fires, volcanic eruptions, dust storms), anthropogenic activities are the main cause of the emission of chemical pollutants into the atmosphere [1]. Most of air chemical pollutants of human origin are produced by the combustion of fossil fuel to produce heat and energy, major industrial processes, exhaust from transportation vehicles (aircraft, cars) and agricultural sources (livestock farms, application of fertilizers, herbicides and pesticides in crop production). Air pollution is composed of a heterogeneous mixture of compounds, categorized into two broad groups: primary and secondary pollutants [2]. Primary pollutants are emitted directly from pollution sources, and include gases ($CO_2$, CO, $SO_2$, NO, $NO_2$), low molecular weight hydrocarbons, persistent organic pollutants (e.g., dioxins), heavy metals (e.g., lead, mercury) and particulate matter (PM). Secondary pollutants are formed in the atmosphere through chemical and photochemical reactions involving primary pollutants; they include ozone ($O_3$), $NO_2$, peroxy acetyl nitrate, hydrogen peroxide and aldehydes [1,2]. Gaseous pollutants are mainly produced by fuel combustion (CO from incomplete combustion and $SO_2$ from combustion of sulfur-rich fuels), while dioxines are produced when materials containing chlorine are burned. Airborne particulate matter (PM) is a major concern especially in the air of densely populated urban areas; it consists of mixtures of particles of different size and composition. Depending on their aerodynamic diameter, they are commonly referred to as $PM_{10}$ (<10 μm), $PM_{2.5-10}$ (coarse particles, 2.5–10 μm), $PM_{2.5}$ (fine particles, <2.5 μm) and ultrafine particles (UFP, <100 nm) [3]. The composition of PM varies, because they can absorb and carry on their surface a great variety of pollutants, such as gases, heavy metals,

organic compounds, polyaromatic hydrocarbons, directly related to their toxicity. In countries such as Northern India and China, particularly high levels of $PM_{2.5}$ are detectable (annual average over 50 μg/m$^3$) [4], subject to seasonal fluctuations and higher than the World Health Organization (WHO) recommendations [5].

## 2. Effects of Air Pollution on Human Health

Exposure to air pollution is associated with increasing morbidity and mortality worldwide. Airborne pollutants may penetrate the human body through multiple routes, including direct inhalation and ingestion, as well as dermal contact, and they cause well-documented acute and long-term effects on human health. Once inhaled, airborne pollutants can affect respiratory system, with airways irritation, bronchoconstriction and dyspnoea, lung inflammation and worsening of conditions of patients with lung diseases [1]. Epidemiological and clinical studies have shown that air pollution is also associated with cardiovascular diseases, and a relationship of exposure to air pollutants with the risk of acute myocardial infarction, stroke, ischaemic heart disease and increase in blood pressure was reported [6]. Moreover, there is increasing evidence that outdoor pollution may have a significant impact on central nervous system and may be associated with some neurological diseases, such as Alzheimer's disease, Parkinson's disease and neurodevelopmental disorders [7]. Air pollution is also considered a risk factor in the incidence of some other pathological conditions, such as autism, retinopathy, low birth weight and immunological dysfunctions [8].

## 3. Pollution-Induced Skin Damage

Being the largest organ of the human body as well as the boundary between the environment and the organism, the skin unsurprisingly is one of the major targets of air pollutants. Air pollution has considerable effects on the human skin, and it is generally accepted that every single pollutant has a different toxicological impact on it. Recently, many Authors reported potential explanations for outdoor air pollutants impact on skin damaging, focusing their interest especially on PM and ozone.

PM are essentially combustion particles formed by a core of elemental carbon coated with a variety of chemicals, such as metals, organic compounds, particularly polycyclic aromatic hydrocarbons (PAHs), nitrates and sulfates [9]. PM induce in skin oxidative stress, producing reactive oxygen species (ROS) and causing the secretion of pro-inflammatory cytokines (TNF-$\alpha$, IL-1$\alpha$, IL-8) [10]. As a consequence of the increased production of ROS, an increase of matrix metalloproteinases (MMPs) occurs, resulting in the degradation of mature dermal collagen, which contributes to skin aging [11]. Coarse PM produce ROS essentially through transition metals (iron, copper, vanadium, chromium) absorbed on their surface, which are able to generate ROS (especially OH$^\circ$) in the Fenton's reaction, while smaller particles produce ROS due essentially to the presence of PAHs and quinones [12]. Quinones are by-products of diesel fuel combustion, but can also be produced in the skin through biotransformation of PAHs by some enzymes (cytochrome P450, epoxide hydrolase and dihydrodiol dehydrogenase) [11]. Li et al. (2003) [13] demonstrated that ultrafine particles had the highest ROS activity compared to coarse and fine particles. PAHs are highly lipophilic carbon compounds with two or more fused aromatic rings, emitted to the atmosphere primarily from the incomplete combustion of organic matter. PAHs absorbed on the surface of airborne PM can penetrate into intact skin and exert direct effects on epidermis cells, such as keratinocytes and melanocytes [2]. PAHs are ligands for the aryl hydrocarbon receptor (AhR), a ubiquitous ligand-dependent cytosolic transcription factor. When AhR ligands engage the receptor, a conformational change occurs in it, which leads to its nuclear translocation and subsequent binding and activation of several genes, included genes encoding several phase I and II xenobiotic metabolizing enzymes (e.g., cytochrome P4501A1, glutathione-*S*-transferase) [14]. The oxidized products of PAHs metabolized by these enzymes induce oxidative stress responses in cells and confirm the involvement of PAHs in the genesis of skin damage due to air pollution. Pan et al. [15] explored the effect of PM on the function of skin

barrier, and showed that particulate matter disrupt stratum corneum and tight junctions both in in vitro and in vivo experiments in pigs, also promoting the skin uptake of some drugs.

Ozone occurs in the stratosphere (where it acts as a filtering barrier to UVC and partly UVB and UVA radiations) and in the troposphere, where it is present as a main component of photochemical smog [16]. At ground-level it is normally found in low concentrations, but it can be formed in higher amounts through interaction of UV radiations with hydrocarbons, volatile organic compounds (VOCs) and nitrogen oxides, becoming a ubiquitous pollutant in the urban environment with concentrations ranging from 0.2 to 1.2 ppm [17]. Due to its peculiar anatomical position, skin is one of the tissues more exposed to the detrimental effects of ozone, especially during smoggy and $O_3$-alert days [18]. Although Afaq et al. [19] showed in human epidermal keratinocytes that AhR is an ozone sensor in human skin, suggesting that AhR signalling is an integral part of induction of cytochrome P450 isoforms by $O_3$, ozone should not reach viable skin cells due to its high reactivity, and it is common opinion that its main cutaneous target is the stratum corneum [2]. Ozone represents an important source of oxidative stress for skin; studies on animal models showed that ozone exposure leads to a progressive depletion of vitamin E and hydrophilic antioxidants (urate, ascorbate, glutathione) and to malondialdehyde (MDA) production in murine stratum corneum [20,21], inducing oxidative damage to lipids and leading to a perturbation of epidermal barrier function. A study performed on the skin of human volunteers showed that the ozone exposure significantly reduced vitamin E levels and increased lipid hydroperoxides in the stratum corneum, confirming that the effects of $O_3$ are limited to the superficial layers of the human skin [22]. In addition to increasing oxidative stress and decreasing of antioxidant skin defenses, ozone exposure is able to induce pro-inflammatory markers and increase the levels of heat shock proteins in mice skin [23]. Inflammatory reactions in turn induce the production of ROS, thus triggering a vicious circle [24]. Detrimental effects of $O_3$ on skin can be enhanced by simultaneous exposure to UV radiation [24]. The use of topical antioxidant mixtures has proven to be effective in preventing $O_3$-induced oxidative damage both in human keratinocytes in culture [18] and in reconstructed human epidermis [25].

Air pollution, with other exogenous factors such as UV radiation and smoking, is definitely recognized as an important extrinsic skin-aging factor, whose pivotal mechanism is the formation of ROS and the subsequent oxidative stress, which can trigger further cellular responses. The skin is equipped with an elaborate antioxidant defense system including enzymatic and nonenzymatic, hydrophilic and lipophilic elements; however, when the extent of the oxidative stress exceeds skin's antioxidant capacity, it leads to oxidative damage, premature skin aging and eventually skin cancer [26].

Until today, no standard protocol is available to objectively substantiate the "anti-pollution" claim, though several in vitro and in vivo tests have been proposed to this purpose. In vitro tests are based on the use of cell cultures or reconstituted skin models and evaluate several biomarkers (PGE2, IL-1$\alpha$, MDA, superoxide dismutase, catalase, gluthathione reductase, to name just a few) after pollutants exposure [27,28]. In vivo tests, performed on volunteers' panels, include instrumental evaluation of skin parameters (such as skin hydration, transepidermal water loss or TEWL, skin elasticity, wrinkles, skin pigmentation) and the evaluation of the levels of oxidative stress and inflammatory markers after exposure to pollutant stressors [29,30]. The availability of reliable and specific markers of airborne pollution upon skin would allow to evaluate and quantify the cutaneous impact of this phenomenon, as well as to assess the effectiveness of ingredients or finished products in counteracting detrimental effects of air pollutants. Recently, the oxidation of squalene has been recognized as a useful model [31]. Squalene, a high-unsaturated triterpene produced by human sebaceous glands and present in sebum with an average concentration of 12%, is very prone to oxidation and is one of the main targets of oxidative stress induced by air pollution. Its peroxidized by-products are considered inflammatory mediators and are involved in comedogenesis, acne and wrinkles formation [31]. Pham et al. [31] established various protocols to evaluate the influence of different pollutants upon squalene oxidation by determining the amount of squalene oxides produced, and concluded that squalene oxidation is a reliable marker of pollution-induced skin damage.

Recently, a small number of studies investigated the cause-effect relationship between air pollution and skin quality. Vierkötter et al. [29] found a significant association between traffic-related airborne particles and extrinsic skin aging signs (pigment spots and wrinkles) in a group of German women. In particular, an increase in soot (per $0.5 \times 10^{-5}$ per m) and particles from traffic (per 475 kg per year and square km) were associated with 20% more pigment spots on forehead and cheeks [29]. The influence of air pollutants on a number of skin parameters was evaluated in a clinical comparative study conducted on 96 subjects in Mexico City (one of the world's most polluted cities) and 93 subjects in Cuernavaca, considered a town preserved from urban pollution [30]. In this comparative study, the Authors studied quantitative and qualitative modifications of a number of skin parameters. The results of this investigation demonstrated that moisturizing was significantly higher in Cuernavaca population; an increased level of sebum excretion rate, a lower level of vitamin E and squalene (the main antioxidants at the surface of the skin) in sebum, and an increase of lactic acid and a higher erythematous index of the face of subjects were documented in Mexico City group. In the stratum corneum a higher level of carbonylated proteins, a lower level of IL $1\alpha$, a decrease of ATP concentration and a decrease of chymotrypsin like activity were detected. A clinical evaluation conducted by dermatologists on the same groups showed a general tendency of a higher incidence of skin problems (skin urticaria, atopy disease, hand dermatitis, dermographism) in Mexico City population compared to Cuernavaca population.

In addition to the effects on healthy skin, a number of studies have shown that outdoor air pollution is a relevant risk factor for the development of atopic dermatitis, a chronic inflammatory skin disease, and can also exacerbate this condition [32–34]. As a consequence, the prevalence of atopic dermatitis in urban areas is significantly higher compared to that of rural areas [34].

## 4. Cosmetic Strategies for Anti-Pollution Skin Defense

The awareness of detrimental effects of environmental pollutants on skin has increased enormously in the most recent years not only within the scientific community but also among consumers. As a consequence, the anti-pollution trend, originated in Asia (home to some of world's most polluted urban areas) and subsequently spread to Western markets, is nowadays a rising trend in cosmetics and personal care industry worldwide, and cosmetic brands are unceasingly developing new concepts and new active ingredients to meet consumers' demand. Several cosmetic strategies can be adopted to protect human skin against environmental pollution. The very first step in an effective cosmetic anti-pollution routine is a proper cleansing of the skin to remove chemicals deposited on it. Another way to defend the skin against environmental stressors is the isolation of the epidermis through the formation of a cohesive and non-occlusive film on its surface, preventing the direct contact with airborne pollutants; this physical barrier can be obtained through the use of film-forming ingredients, both synthetic (silicones, acrylic acid copolymers) and naturally derived (peptides and polysaccharides extracted from plants or obtained by fermentation processes). The third approach is the inclusion in anti-pollution formulations of antioxidants, in order to protect against free radical effects, or ingredients able to up-regulate the antioxidant defenses of the epidermis cells [2]. Some cosmetic companies introduce in their anti-pollution cosmetics several ingredients with different complementary mechanisms of action, obtaining formulations designed to tackle as many pollutants as possible. Figure 1 presents schematically the action of environmental pollutants on human skin and the main mechanisms of antipollution cosmetic ingredients.

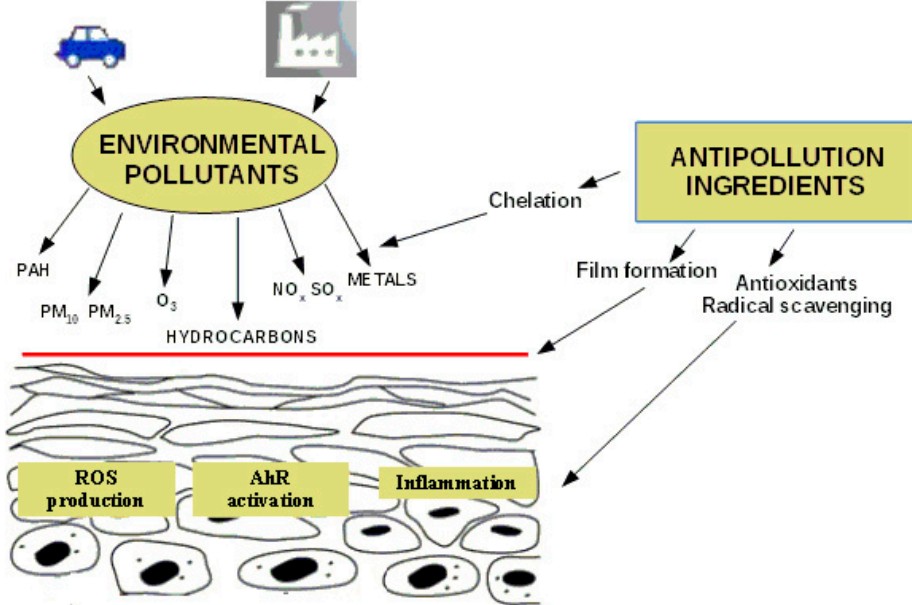

**Figure 1.** Mechanisms of the effects of environmental pollutants on skin and strategies for anti-pollution skin defense.

Most of the active anti-pollution ingredients present in formulations on the market are products of botanical origin. This reflects a more general trend in the today's cosmetics and personal care industry. Indeed plants contain countless metabolites with potential cosmetic applications, which combine efficiency, reduced risk of irritation and allergies, reduced adverse effects and the possibility to refer on the labels of beauty products placed on the market to claims such as "organic", "environmental sustainability" and "fair trade"; these claims are increasingly popular amongst the consumers due to the ever-increasing demand for more ethical, natural and "green" formulations [35].

The current review aims to present a selection of the most popular ingredients of botanical origin marketed by suppliers with the claim "anti-pollution", and the scientific rationale behind their cosmetic applications. A number of commercially available anti-pollution cosmetic ingredients of botanical origin were retrieved by an electronic survey conducted by the popular search engine Google and the technical websites for chemicals and materials Prospector, SpecialChem and Cosmetic Design Europe, using the key words "anti-pollution cosmetics" and "anti-pollution ingredients". For practical reasons, it has been decided to include in this review only the ingredients derived from a single botanical species, thus excluding the products containing mixtures of plant extracts. For each ingredient the technical documentation (product brochure, datasheet, experimental data substantiating the claims) was acquired from the manufacturers' own websites. Subsequently, scientific papers, found by using the academic search engines Google Scholar, ScienceDirect and PubMed, were consulted to verify the scientific soundness of the anti-pollution claims of these botanical extracts plants. For the ease of the readers, the anti-pollution ingredients taken into account in this review were divided into two tables, according to their botanical origin: Algae and Spermatophytae. In these tables, the trade name, the supplier, the INCI name, the supplier claims and the recommended concentrations were reported for each ingredient.

## 5. Anti-Pollution Ingredients from Algae

Marine algae are eukaryotic organisms classified in microalgae (unicellular species present in phytoplankton) and macroalgae (found in coastal areas). Macroalgae (seaweeds) are in turn classified in *Rhodophyceae* (red algae), *Chlorophyceae* (green algae) and *Pheophyceae* (brown algae), according to their dominant pigment [36]. Algae provide a great variety of metabolites (polysaccharides, lipids, phenolic

compounds and pigments) [37] and can be easily cultured on seashores in great volumes; moreover, they grow quickly, and it is possible to control the production of their metabolites by manipulating the culture conditions [36]. For all these reasons, algae represent an attractive renewable source of bioactive compounds with potential applications in pharmaceutical, nutraceutical and cosmetics industries [38].

A number of bioactive compounds and extracts derived from macroalgae have proven to be useful in the treatment of some skin conditions [37]. Some algae species produce bioactive molecules with photo-protective activity due to their ability to absorb UV-A and UV-B radiation [37,38]; other algal species are potential sources of skin whitening agents, since they produce metabolites (e.g., fucoxantin, phloroglucinol) able to inhibit natural tyrosinase [39]. Moreover, some compounds derived from algae exhibit antibacterial and anti-inflammatory activity and can be useful in the management of acne-affected skin [37]. Other bioactivities from algae are closely linked to the use of seaweed-derived products as anti-pollution cosmetic ingredients. In particular, researchers have extensively investigated the antioxidant activity of algae extracts; indeed algae, due to the extreme conditions in which they often live, are naturally exposed to oxidative stress, and develop efficient strategies to protect against the effects of ROS and other oxidizing agents [38]. The antioxidant potential of a variety of algal species extracts was demonstrated with different methods, such as 2,2-diphenyl-1-picrylhydrazyl (DPPH°) free-radical-scavenging, ferric-reducing antioxidant power (FRAP), ABTS (2,2′-Azino-bis(3-ethylbenzothiazoline-6-sulfonic acid) radical scavenging, in vitro copper-induced oxidation of human LDL (Low Density Lipoprotein) assay, reducing activity, metal chelating assay, scavenging ability on hydroxyl and superoxide radicals [40,41]. Brown algae have been reported to contain comparatively higher contents and more active antioxidants than red and green algae [42]. A statistically significant correlation between this antioxidant activity and the total polyphenol content of these extracts was demonstrated [42,43], suggesting that this class of compounds is at least in part responsible for the antioxidant properties of seaweed extracts. Amongst the many polyphenols been identified in algal extracts, of particular interest are phlorotannins, formed by polymerization of phloroglucinol units linked together in different ways. Phlorotannins only exist within brown algae, are not found in terrestrial plants [42], and can be divided into six categories (fucols, phlorethols, fucophlorethols, fuhalols, isofuhalols and eckols). They possess a strong antioxidant activity related to phenol rings in their structure, and having up to eight rings they are more efficient free radical scavengers when compared to polyphenols from terrestrial plants, which have 3–4 rings [44].

Other components which contribute to the antioxidant potential of algae are sulfated polysaccharides, that in recent times have attracted the interest from life science researchers owing to a wide range of biological activities with potential health benefits, such as anti-allergic, anti-HIV, anticancer, anticoagulant and anti-oxidant activities [45]. Sulfated polysaccharides are anionic polymers widespread among marine algae but also occurring in animals; their chemical structure varies depending on the seaweed species that they come from. The most important sulfated polysaccharides recovered in marine algae are ulvans in green algae, carrageenans in red algae and fucoidans and laminarians in brown algae [46].

Ulvans are water-soluble sulfated heteropolysaccharides, mainly constituted by disaccharide repeated units formed by D-glucuronic or L-iduronic acid linked to L-rhamnose-3-sulfate [47]. These polymers exhibit a broad range of biological activities, a notable example being the antioxidant one [47–49]. The antioxidant properties of ulvans are influenced by the extraction procedures and depend on the carbohydrate composition and the sulfate content, since ulvans with higher sulfate content show a significantly higher antioxidant activity [47,49].

The composition, structures and biological properties of fucoidans have been extensively reviewed ([42] and literature therein). They are sulfated polysaccharides found exclusively in the cell walls of brown algae; their major components are L-fucose and sulfate. They consist of $\alpha$-(1→3)- and (1→4)-linked-L-fucopyranosyl residues, organized in stretches of (1→3)-$\alpha$-fucan or of alternating $\alpha$-(1→3)- and $\alpha$-(1→4)-bonded L-fucose residues [50]. Fucoidans exhibit a broad spectrum of biological

activities, including anticancer, apoptosis-inducing, immunomodulatory, antiviral, anti-thrombotic, anti-inflammatory and antioxidant activities [51]. In vitro antioxidant activity of fucoidans has been determined by various methods, such as DPPH free radical scavenging assay, iron-chelating activity assessment, superoxide anion and hydroxyl radical scavenging activity, reducing power assay [50,52–54]. Recently, effective methods of extraction of fucoidans (microwave-assisted extraction, high pressure homogenization combined with hydrothermal extraction), alternative to classical extraction methods (hot water, diluted acid, diluted alkali), time-expensive and associated to multi-step processes, high temperature, large solvent volumes, were developed [50,53]. Fucoidans extracted by these techniques exhibited in some cases a higher antioxidant activity than those extracted by conventional methods [53]. In addition to their antioxidant properties, fucoidans possess another biological activity relevant in the cosmetic field: they are able to prevent UVB-induced matrix metalloproteinase-1 (MMP-1) expression and suppress MMP-3 in vitro [39,55]. MMPs induce degradation of dermal proteins such as collagen, fibronectin and elastin, contributing to skin damage; therefore fucoidan may be useful to prevent skin photoaging not only by scavenging ROS formed during exposition to UV radiations, but also by inhibiting the formation of MMPs.

As shown in Table S1, a number of cosmetic ingredients based on algal extracts have been developed and are proposed by manufacturers as functional substances suitable for anti-pollution skincare products. Their use is substantiated not only by the general literature referred so far, but often also, when available, by investigations focused on specific algal species.

The ingredient No. 1 (Table S1), Contacticel™, contains an extract of *Acrochaetium moniliforme*, an epiphytic red macroalga made of cell filaments found in very low quantities in the ocean; the patented Celebrity™ technology produces biomass of this red alga in photobioreactors in a sufficient quantity, unavailable in the sea. Scientific literature on possible biological activities of this alga was not found; the information leaflet of the manufacturer claims that the patented commercial extract limits in vivo excessive sebum production and reduces the ozonolyzed squalene (tests performed versus placebo on two groups of 20 women each, in Shanghai's polluted atmosphere). Moreover, in an in vitro sebocyte model exposed to urban dust the extract proved to be effective in regulating the lipid production.

The antioxidant activity of the edible brown seaweed *Laminaria digitata* (ingredient No. 2, Table S1) is documented by a number of investigations [43,56–59]. The study of Heffernan et al. [43] showed that the crude extracts of *L. digitata* showed a total phenolic content and an antioxidant activity lower than other macroalgae examined, but these parameters improved when the extracts were fractionated with suitable dialysis membranes. Moreover, it has been demonstrated that the thermal treatment increased its content in antioxidant compounds and improved its free radical scavenging activity [58].

The ingredients No. 3 and No. 7 contain as an active anti-pollution ingredient *Undaria pinnatifida* extract; this brown alga is widely used as food and as a remedy in traditional Chinese medicine for over 1000 years. As stated previously for brown algae, *U. pinnatifida* contains sulfated polysaccharides that exhibit good antioxidant activities, related with their sulfate content [54,60]. Moreover, it also contains fucoxanthin, a carotenoid present in the chloroplasts, able to counteract oxidative stress by UV radiation [61].

The ingredient No. 4, designed to be used in hair-care products, contains a hydroglycolic extract of the brown seaweed *Pelvetia canaliculata*. Although *P. canaliculata,* like all brown algae, contains fucoidans [62] and phlorotannins and carotenoids, able to absorb UV radiation and to fight photoxidative stress [63], the leaflet supplied by the manufacturer emphasises the effectiveness of its extract in reducing residues and depositions caused by the action of pollutants, chlorine and the build-up effect of cationic hair conditioners. Alginates and fucoidans contained in the cells walls of *P. canaliculata* are poly-anions due to the presence of carboxylic and sulfonic groups, and as a consequence this alga can act as a natural cation exchanger. This ability is widely documented in the scientific literature, and it was proposed to use *P. canaliculata* biomass to sequestrate and remove metal ions (zinc, iron, copper, trivalent chromium, lead, nickel) from industrial wastewaters [64–66].

All of these biological activities make *P. canaliculata* extract a good candidate for the formulation of anti-pollution cosmetics.

The ingredient No. 5 consist of an extract of *Ascophyllum nodosum,* harvested on Ouessant Island (Brittany) by a hand cutting harvest method; it is concentrated in high molecular weight fucoidans (over 100 kDa), as declared by manufacturer. The antioxidant activity of this brown seaweed is well documented [50,59,67] and can be attributed to the presence of both sulfated polysaccharides and phenolic compounds. *A. nodosum* contains the abovementioned fucoidan but also ascophyllan, another sulfated polysaccharide structurally similar to fucoidan characterized by a more pronounced antioxidant activity, in addition to a wide variety of interesting biological activities [68]. Moreover, *A. nodosum* produces a variety of phenolic compounds, namely phlorotannins, flavonoids and phenolic acid derivatives [69,70]. On the leaflet of the manufacturer the extract No. 5, at the concentration of 3%, is claimed to decrease AhR receptor expression by 73% compared to a placebo in an ex vivo test (human skin explants). Moreover, to this ingredient is ascribed the ability to reinforce the skin barrier; in particular, it is reported that on the model of reconstructed epidermal skin Episkin™ native fucoidans (5%) increased the number of mature corneocytes by 225%, while the whole extract (3%), after 56 days, decreased TEWL, an indicator of the barrier dysfunction, by 13% versus placebo in an in vivo test [71].

Finally, the ingredient No. 6 contains an extract of the green alga *Ulva lactuca*, also known by the common name of sea lettuce and rich in the sulfated polysaccharides ulvans, as mentioned above. If administered orally, *U. lactuca* extracts show anti-inflammatory effect in carrageenan-induced paw oedema in rats [72] and are able to ameliorate hepatic enzymatic and non-enzymatic antioxidant defenses of hypercholesterolemic rats [73].

## 6. Anti-Pollution Ingredients from Spermatophytae

### 6.1. Eriodictyon Californicum

*Eriodictyon californicum* (also known as "yerba santa" and "bear weed") is an evergreen shrub within the *Boraginaceae* family, native to Central America (Mexico and South-West of USA). For centuries Native Americans used it as a medicinal plant to treat several respiratory conditions and skin wounds. The leaves of *E. californicum* are covered by a resin containing flavonoids (such as eriodictyol and homoeriodictyol) [74,75], which act as herbivore deterrents and UV screens; this plant is also a source of moisturizing compounds such as mucopolysaccharides and glycoproteins, which produce their moisturizing effects via hydrogen bonding of water by their sugar moieties.

Extracts of *E. californicum* represent the active substance in ingredients No. 1 and No. 8 in Table S2. Ingredient No. 8 (Phytessence Holyherb) is an extract of flowers, leaves and stems of *E. californicum*, while ingredient No. 1 (ABC Yerba Santa Glycoprotein PF) is obtained by fermenting the leaves of *E. californicum* with the bacterium *Lactobacillus lactis*; this extract obtained is rich in glycoproteins and exhibits moisturizing and soothing properties. In the technical data sheet provided by the manufacturer a remarkable improvement of skin moisturization (30%) produced by Yerba Santa Glycoprotein PF (5%) was reported, whereas in the same experimental conditions *Aloe vera* gel 10× produced an increase of 20%. Ingredient No. 1 was also tested to verify its anti-pollution properties. The extract was applied to the skin, which was then contaminated with a known amount of activated charcoal (>2.5 μm size particles). After washing with a controlled volume of water, the amount of microparticles remained on the skin was evaluated; when compared with an untreated control, the extract was able to prevent the deposition of PM particles into the skin fine lines and wrinkles. The extracts of *E. californicum,* due to the presence of flavonoids such as homoeriodictyol and eriodictyol, well known for their antioxidant activity [76,77], provide further benefits when added to cosmetic formulations.

## 6.2. Camellia Sinensis

The ingredient No. 2 of Table S2 (Berkemyol® Thé vert) is composed by polyphenols extracted from green tea leaves and esterified with palmitic acid. Green tea is obtained by roasting or steaming *Camellia sinensis* (*Theaceae*) leaves in order to inactivate polyphenol oxidase activity. Green tea extracts are complex mixtures of bioactive compounds, including tea polyphenols, primarily green tea catechins, that account for 30–40% of the extractable solid of dried green tea leaves [78]. Tea catechins include epicatechin, epicatechin-3-gallate, epigallocatechin and epigallocatechin gallate [78]. These polyphenols have gained interest in recent years because of interesting biological activities, including antimicrobial, anti-inflammatory, anticancer, antioxidant and radical scavenging activities [79–82]. On the basis of their biological properties, green tea polyphenols are generally accepted as having a protective effect against oxidative stress and DNA and cell structures damage induced by a number of environmental toxins/toxicants (pesticides, smoking, mycotoxins, PCB, arsenic [78]); these properties provide the rationale for the use of green tea extracts as functional ingredients of anti-pollution cosmetics. However, it is known that polyphenols in their native form are unstable because they are susceptible to oxidation induced by several environmental agents (metal element traces, light, oxygen) [83]. Moreover, green tea polyphenols are soluble in water and therefore difficult to use in cosmetic formulations when lipophilic ingredients are required. A way of stabilizing these polyphenols and imparting them lipophilic properties is protecting the phenol functions as fatty acid esters with a method described in the American Patent US 5808119 [84]. To evaluate whether the biological properties of polyphenols are maintained after esterification, studies were performed by using cutaneous explants from abdominoplasty surgery as a model of human skin. After topical application of the green tea extract, free radical production was induced by UV radiation; the lipid peroxidation process was studied by determining the levels of malonyldialdehyde (MDA) as indicator [84]. Stabilized polyphenols showed a good anti-lipoperoxidant activity, higher than that of Vitamin E. Since the radical scavenging of polyphenols related is to the free phenolic OH, it is assumed that esterified polyphenols permeate the skin barrier and then are hydrolyzed by skin esterases to the active forms [85]. Moreover, green tea polyphenols at concentration of 0.1% and 0.25% respectively induce an increase of 18% and 40% of collagen IV in the dermo-epidermal junction; at 0.25% and 0.5% they increase of 13% and 21% fibriline-1 in the dermo-epidermal junction [84].

## 6.3. Marrubium Vulgare

*Marrubium vulgare* is a plant widely used in antipollution skincare products; four ingredients examined in this review (ingredients No. 4, 6, 7 and 8, Table S2) contain *M. vulgare* extracts. The genus *Marrubium* (*Lamiaceae*) includes about 40 species of flowering plants indigenous in Europe, Mediterranean area and Asia. Many species of *Marrubium* are reported in the literature to be used in folk medicine and their extracts have been investigated for their chemical composition and for their antioxidant and lightening properties [86–88]. *M. vulgare* (white horehound), widely used in traditional medicine in some countries, is the most investigated species of *Marrubium*. Its aerial parts are official in Hungarian Pharmacopoeia VII [89], and the European Medicine Agency (EMA) approved in 2013 the treatment of cough associated to cold, mild dyspeptic complaints and temporary loss of appetite as indication for aerial part of this plant [90]. *M. vulgare* is reported to possess several biological activities, among which the most interesting are antihepatotoxic [91], antihyperglicemic [92], antibacterial [93], anti-inflammatory [94] and antioxidant properties [95]. In particular antioxidant properties may justify the widespread of *Marrubium vulgare* extracts, often prepared with peculiar extraction techniques, as natural cosmetic ingredients, with claims including anti-pollution, antioxidant, protective for irritated and stressed skin, detoxifying, soothing. Reported results suggest a remarkable antioxidant activity of *M. vulgare* extracts assessed with different methods (DPPH° radical scavenging, scavenging activity against hydrogen peroxide, iron reducing power) [95–97]. In order to correlate this antioxidant activity to specific bioactive compounds, phytochemical composition of *M. vulgare* has been extensively investigated; these studies, conducted on different types of extracts, led to the identification of a wide

array of phytochemical compounds, such as flavonoids, terpenoids and phenylethanoid glycosides. Several flavonoids were isolated from *M. vulgare*, including luteolin, apigenin, terniflorin, anisofolin A, ladanein [95,98]. Phytochemical screening revealed the presence of several terpenoid compounds, such as marrubiin, premarrubiin, marrubenol, sacranoside A, deacetylforskolin, preleosibirin, marrulibacetal [95,99]. Finally, several phenylpropanoid compounds were isolated and identified from *M. vulgare*: caffeoyl-L-malic acid, verbascoside, decaffeoylverbascoside, forsytoside B, alyssonoside, leukoceptoside A, acteoside, arenarioside, ballotetroside [95,100]. Ladanein, verbascoside and forsythoside B showed a relevant antioxidant activity in vitro [98]; on the other hand, experimental data suggest that natural phenylpropanoids could protect cells from oxidative stress [101,102]. These studies justify the use of *M. vulgare* extracts as cosmetic ingredient and support scientific substantiation of the anti-pollution claim.

### 6.4. Schinus Molle

*Schinus molle* (*Anacardiaceae*), also known as Peruvian pepper tree, false pepper or pink pepper, is an evergreen tree native to Peruvian Andes. Widely used in traditional medicine for its purported analgesic, antidepressant, antimicrobial, diuretic, astringent and antispasmodic properties, *Schinus molle* exhibits insect repellent, anti-inflammatory, antifungal and antioxidant effects [103]. The antioxidant properties of leaf and fruit essentials oils of *S. molle* were demonstrated by using DPPH° free radical, ABTS and β-carotene/linoleic acid methods [104,105]. Methanolic extracts of bark and flowers of *S. molle* were also tested for their DPPH° scavenging activity, and they exhibited a remarkable antioxidant activity when compared to quercetin [106]. In aqueous methanolic extracts of the leaves of *S. molle* a number of polyphenolic metabolites were found, including glycosides based on quercetin as an aglycone [107]. Some of them exhibited moderate to strong radical scavenging properties on lipid peroxidation, OH° and superoxide anion generation; the most active compounds were miquelianin and quercetin 3-*O*-β-D-galacturonopyranoside [107]. An extract of *Schinus molle* (Elixiance™, product No. 4, Table S2) is marketed as a cosmetic ingredient with anti-pollution, anti-aging and anti-wrinkle benefits. In the technical datasheet provided by the manufacturer it is stated that Elixiance™ is rich in polyphenols such as quercitrin and miquelianin. As far as the anti-pollution benefits of this extract are concerned, the document claims that it "...limits the effects of air pollution in vitro ($PM_{2.5} = -11\%$; $PM_{10} = -38.5\%$) ... ", " ... contributes to reduction in skin permeability induced by environmental stress (in vitro) ... " and " ... is associated with anti-pollution benefits supported by a clinical study on 39 volunteers in Shangai." Moreover, a skin-purifying effect, characterized by a reduced quantity of skin sebum and by a decrease of the appearance of pores, is associated to *S. molle* extract.

### 6.5. Camellia Japonica

*Camellia japonica*, also known as Rose of winter, is a flowering tree or shrub belonging to the Theaceae family and naturally occurring in China, Japan and Korea. An extract of *C. japonica* flowers is the active component of the ingredient No. 9 of Table S2 (RedSnow®). It has been reported that *C. japonica*, whose flowers and flower buds were traditionally used in oriental medicine as an astringent, anti-hemorrhagic and anti-inflammatory remedy, exhibits a variety of biological activities, such as antiviral, anti-atherogenic, anti-hyperuricemic, anti-photoaging, antioxidant, radical scavenging and anti-inflammatory effects, and glycation inhibitory action [108–112]. The ethanol extract of *C. japonica* flowers exhibits antioxidant properties by scavenging ROS (superoxide and hydroxyl radicals) in a free-cell system and in human HaCaT keratinocytes; moreover, it is able to increase the protein expression of the antioxidant enzymes superoxide dismutase, catalase and glutathione peroxidase [108]. The ROS scavenging effect and the induction of antioxidant enzymes of *C. japonica* extract may be associated with the presence of antioxidant phenolic compounds such as quercetin and kaempferol glycosides [108,113]. In a study on the anti-aging properties of *C. japonica* flower extract in an ex vivo model, it has been shown that it reduces piknotic nuclei and it prevents the detachment of the

dermo-epidermal junction induced by pollutants such as heavy metals and hydrocarbons; moreover, it also induces an increase of collagen I and a decrease of MMP-1 [114]. These results support the use of *C. japonica* flower extract in anti-aging and anti-pollution cosmetics.

### 6.6. Schisandra Chinensis

*Schisandra chinensis* Baill (*Schisandraceae*) is a plant native to China, Japan and Russia; its dried fruits are used in traditional Chinese medicine, where it is considered one of the 50 fundamental herbs. Modern studies show that this plant possesses several biological activities such as anti-hepatotoxic, antitumour, anti-inflammatory and antioxidant [115]. The major constituents of the fruit extract of *S. chinensis* are lignans, a large group of naturally occurring phenols classified into several group according to their chemical structure. The fruit extract of *S. chinensis* (but also leaves and stems) especially contains lignans with dibenzocyclooctadiene skeleton, such as schisandrol, schisantherin A, deoxyschisandrin, schisandrin B and schisandrin C [116], and with tetrahydrofuran structure, such as D-epigalbacin, machilin G and chicanine [117]; several scientific studies demonstrated the antioxidant activity of these compounds [115,117,118]. Some recent studies have also shown the presence in fruits and leaves of *S. chinensis* of phenolic compounds with good antioxidant activity, such as chlorogenic acid, isoquercitrin and quercitrin [115,119]. Schisandrin also exhibits anti-inflammatory activity [120,121], and *S. chinensis* fruit extract has been reported to reduce pro-inflammatory cytokine levels in THP-1 cells stimulated with *P. acnes* and to protect UVB-exposed fibroblasts from photoaging [122]. Due to these biological properties, the extract from *S. chinensis* fruits has beneficial effects on the skin [118], and has been proposed as a cosmetic anti-pollution ingredient (Table S2, Ingredient No. 10, Urbalys®). This ingredient contains schisandrin 8–12% (by dry matter). In the datasheet provided by the manufacturer it is claimed that this product is able to reduce the NQ01 (NAD(P)H dehydrogenase 1) expression, to limit the expression of MT1H (Metallothionein 1H) and to protect from inflammation on a 3D reconstructed full-thickness skin model exposed to a mixture of urban pollutants (in vitro tests); in vivo tests performed on female volunteers exposed to urban pollution showed a conservation of basal TEWL, an amelioration of skin radiance and an improvement of microcirculation and tissular oxygenation [123].

## 7. Conclusions

Several scientific investigations have established that the prolonged exposure to environmental pollutants can produce in human skin biochemical parameters modifications and impairment of barrier function, and can promote the mechanisms of skin aging; the visible results of these effects are dryness, wrinkles, dark spots, sagging and the aggravation of skin sensitivity. As the awareness of the impact of environmental stressors on the skin grows, there is an increasing consumer demand for cosmetics and personal care products able to provide anti-pollution benefits. The anti-pollution skincare is one of the latest cosmetic trends; started in Asia, it is currently gaining ground all over the world, and new solutions, ingredients and products specifically designed to offer skin protection against pollution are continuously developed. With the growth in demand for natural cosmetics steadily on the rise, it is natural that plant extracts are becoming the most popular ingredients of cosmetics designed to fight skin pollution; indeed plant extracts are often rich in bioactive compounds whose activities (antioxidant, chelating, film-forming) can be exploited in anti-pollution formulations. As stated above, airborne pollutants induce adverse effects on human skin mainly via oxidative damage, with a consequent oxidative stress and a depletion of ant ioxidant enzymes and other antioxidant substances in epidermis. For this reason, it is not surprising that most of the plants used as a source of anti-pollution cosmetic ingredients contain antioxidants as active substances.

This review was aimed to give a representative list of the most popular anti-pollution cosmetic ingredients of botanical origin, describe their mechanism(s) of action and provide a scientific rationale justifying their use. This list is not exhaustive; indeed, manufacturers are expected to propose an

increasing number of plant derivatives as active ingredients of antipollution cosmetics, since the demand for this skincare segment is here to stay and it will even increase.

**Supplementary Materials:** The following are available online at http://www.mdpi.com/2079-9284/5/1/19/s1, Table S1: Anti-pollution cosmetic ingredients obtained from Algae, Table S2: Anti-pollution cosmetic ingredients obtained from Spermatophytae (n.s. = not specified).

**Acknowledgments:** This work was partially supported by MIUR (Ministero dell'Istruzione, dell'Università e della Ricerca), Italy.

**Author Contributions:** Authors have contributed to the literature search and to the preparation of the review in equal measure.

**Conflicts of Interest:** The authors declare no conflict of interest.

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
