# Peer review of "Cosmetic Functional Ingredients from Botanical Sources for Anti-Pollution Skincare Products"

_cosmetics, doi:10.3390/cosmetics5010019_

Round 1
Reviewer 1 Report
Dear Authors,
What is [7 and literature therein]?
The manuscript is of great scientific interest. The manuscript is written clearly and concisely. In my opinion, this manuscript “Cosmetic Functional Ingredients from Botanical Sources for Anti-Pollution Skincare Products ” can be published in this form.
Author Response
Dear Referee
Thank you very much for your appreciation and for your positive comments.
- On page 2, line 61 of our manuscript (corresponding to line 71 of the revised manuscript) we have indicated (in red bold characters) the bibliographic references cited in reference 8 (7 in the non revised version of the manuscript) describing the relationship between air pollution and autism, retinopathy, low birth weight and immunological dysfunctions.
My best regards
Claudia Juliano
Reviewer 2 Report
The paper is well written and clearly demonstrates the potential use of plant-derived products to treat cosmetic problems associated with pollution. This review in the present form makes an excellent snapshot of the actual use of plants in cosmetics.
My only reccomendation is to perform minor spell and grammatical corrections, which not affect the quality of the work.
Author Response
Dear Referee
Thank you very much for your appreciation and for your positive comments.
My best regards
Claudia Juliano
Reviewer 3 Report
This review manuscript describes some commercial cosmetic ingredients obtained from botanical sources able to reduce the impact of air pollutants on human skin. The authors discuss the different mechanisms involved. The manuscript is well organized and carefully written and provides valuable information, suitable for the cosmetic industry.
1) In the abstract, give a few details on the composition/toxicity of particulate matter.
2) Line 229-230: The authors stated that ‘In general, brown algae present a higher antioxidant activity compared to red and green algae’. This is not a rule and depends on a particular algae strain. Please revise.
3) The authors may consider adding a picture summarizing the strategies for anti-pollution skin defense.
4) The relationship between cosmetic ingredients, anti-pollutant effect and antioxidant enzymes (e.g. superoxide dismutase) should be discussed.
Author Response
Dear Referee
Thank you very much for your appreciation and for your positive comments.
In the abstract, a short sentence in brackets (line 11-12) was added to give short details about particulate matter.
- Lines 229-230 of the non-revised manuscript (lines 309-311 of the revised manuscript): “In general, brown algae present a higher antioxidant activity compared to red and green algae [42].” We have replaced this sentence with a statement quoted verbatim from Balboa et al.,Food Chem. 2013, 138, 1764-1785: “Brown algae have been reported to contain comparatively higher contents and more active antioxidants than red and green algae [42].”
- A picture summarising the strategies for antipollution skin defense was added to the manuscript and a short sentence to introduce it was added on 223-225 lines of the revised manuscript.
- A sentence has been added (lines 628-632 of the revised manuscript, bold red characters) in “Conclusions” paragraph about the importance of antioxidant activity of antipollution ingredients.
Reviewer 4 Report
Nice article, scientifically sound with an impressive number of relevant references.
General comment:
As the explained impact of airborne pollutions on various parameters of the skin is very much depending on the exposure details the reviewer would like to see some more influencer like: duration, dose, phys.-chem. character of the pollution or occlusion.
More specific:
Line 44: Reference to the original WHO article is recommended.
Line 148: In this context skin moisturizing may be predominantly determined by the specific food uptake, probably more relevant as by the exposure to airborne pollutions.
Line 227: Some remaining abbreviations should be explained like: ABTS, DPPH or LDL.
Author Response
Dear Referee
Thank you very much for your appreciation and for your positive comments.
General comment.
Unfortunately there are only a small number of studies about the impact of air pollution on skin quality, and they report very few numerical data about the exposure details you are interested in. Vierkötter et al. (reference 29) studied two groups of women, one from the Ruhr area and one from the rural area of Borken. The Authors gave some figures about the extent of pollution in those areas: “The mean level of traffic-related particle emission was 899.9 kg a-1